# DIFFUSIONTREND: A MINIMALIST APPROACH TO VIRTUAL FASHION TRY-ON

## ABSTRACT

In this paper, we introduce DiffusionTrend, a pioneering approach for virtual fashion try-on that forgoes the need for training diffusion models, thereby offering simple, conventional pose virtual try-on services with significantly reduced computational overhead. By leveraging advanced diffusion models, DiffusionTrend harnesses latents rich with prior information to capture the nuances of garment details. Throughout the diffusion denoising process, these details are seamlessly integrated into the model image generation, expertly directed by a precise garment mask crafted by a lightweight and compact CNN. Although our DiffusionTrend model initially demonstrates suboptimal metric performance, our exploratory approach offers several significant advantages: (1) It circumvents the need for resource-intensive training of diffusion models on large datasets. (2) It eliminates the necessity for various complex and user-unfriendly model inputs. (3) It delivers a visually compelling virtual try-on experience, underscoring the potential of training-free diffusion models for future research within the community. Overall, this initial foray into the application of untrained diffusion models in virtual try-on technology potentially paves the way for further exploration and refinement in this industrially and academically valuable field.

## 1 INTRODUCTION

Virtual try-on technology (Song et al., 2023; Islam et al., 2024), which digitally superimposes images of models wearing various outfits, represents a pivotal innovation in the fashion sector. This advancement offers consumers an immersive and interactive experience with garments, allowing them to preview how clothing might look on them without physically visiting stores. This not only bolsters consumer confidence in their purchasing decisions but also conserves considerable time and effort. The shopping experience might be elevated due to convenience and efficiency. For retailers, virtual try-on technology streamlines their operations by obviating the need to employ live models for merchandise display. It also circumvents the financial burdens associated with traditional product photography, leading to a marked improvement in operational efficiency. From the perspective of commercial platforms, virtual try-on technology serves as a magnet for attracting a larger user base and fostering user loyalty. Moreover, it facilitates accurate market forecasting and trend analysis through data-driven insights, which in turn, stimulates the commercial growth of the entire platform. Collectively, virtual try-on is reshaping the landscape of the apparel retail industry, propelling the fashion sector towards a future characterized by heightened efficiency and customization.

Traditional virtual try-on solutions (Choi et al., 2021; Lee et al., 2022; Ge et al., 2021; Xie et al., 2023) are predicated on a two-stage pipeline utilizing Generative Adversarial Networks (GANs) (Goodfellow et al., 2014). The initial stage in this framework involves the application of an explicit warp module to deform the clothing to the desired area on the body. The subsequent stage integrates the deformed clothing using a GAN-based try-on generator. To attain accurate clothing deformation, earlier methodologies (Bai et al., 2022; Ge et al., 2021; Han et al., 2019; Lee et al., 2022; Xie et al., 2023) have employed a trainable network designed to estimate a dense flow map (Zhou et al., 2016), thereby facilitating the mapping of the clothing onto the human form. In parallel, various approaches (Choi et al., 2021; Ge et al., 2021; Xie et al., 2023; Lee et al., 2022; Yang et al., 2020; Issenhuth et al., 2020), have been proposed to address the misalignment issues between the warped clothing and the human body. Techniques such as normalization (Choi et al., 2021) and distillation (Ge et al., 2021; Issenhuth et al., 2020) have been implemented to mitigate these discrepancies. Recent advancements

Table 1: Comparison of input requirements for previous virtual try-on models. A checkmark (✓) indicates that the input modality is required, while a dash (-) indicates that it is not or not mentioned.

| Method | Clothes Mask | Densepose | Segment Map | Clothes-Agnostic | Keypoint |
|---|---|---|---|---|---|
| TryOnDiffusion (Zhu et al., 2023) | - | - | ✓ | ✓ | ✓ |
| DCI-VTON (Gou et al., 2023) | ✓ | ✓ | ✓ | ✓ | - |
| LaDI-VTON (Morelli et al., 2023) | ✓ | ✓ | - | ✓ | - |
| WarpDiffusion (Li et al., 2023) | ✓ | - | - | ✓ | - |
| OOTDiffusion (Xu et al., 2024) | ✓ | - | ✓ | - | ✓ |
| StableVITON (Kim et al., 2023) | ✓ | ✓ | - | ✓ | - |
| IDM-VTON (Choi et al., 2024) | ✓ | ✓ | - | ✓ | - |
| Wear-Any-Way (Chen et al., 2024) | ✓ | - | - | ✓ | ✓ |
| DiffusionTrend (Ours) | ✓ | - | - | - | - |

in the field of diffusion models (Ho et al., 2020) have led to a notable enhancement in the quality of multiple of image synthesis tasks, with a particular emphasis on the domain of virtual try-on. In this context, contemporary research endeavors have leveraged pre-trained text-to-image diffusion models to produce high-fidelity results for virtual try-on applications. The TryOnDiffusion model (Zhu et al., 2023), for instance, employs a dual U-Net architecture to adeptly perform the try-on task. LADI-VTON (Morelli et al., 2023) and DCI-VTON (Gou et al., 2023) either conceptualize clothing items as pseudo-words or integrate garments through the use of warping networks into existing pre-trained diffusion models. These innovative techniques have collectively contributed to the evolution of virtual try-on technology, offering more sophisticated and realistic outcomes.

Despite these advancements, a review of existing studies reveals that the high precision and realism achieved by current methods require training on extensive try-on datasets (Morelli et al., 2022), particularly in diffusion-based approaches (Zhu et al., 2023; Morelli et al., 2023; Gou et al., 2023). Indeed, diffusion models contain up to 800 million parameters and simulate the data generation process by iteratively modeling the diffusion and reverse diffusion steps. Each iteration involves complex probability distribution calculations and requires substantial computational resources, leading to considerable overall training costs. In the words, current methods rely on resource-intensive network training. While intensive training is well-suited for generating complex poses and capturing fine image details, the substantial computational costs can make it an unattractive option. This is particularly true since, in most cases, consumers make purchase decisions after a simple try-on with regular poses. Besides, as shown in Table 1, these methods often demand multiple types of model inputs, such as densepose (Güler et al., 2018), segmentation maps (Gong et al., 2017), clothes-agnostic images/masks/representations (Han et al., 2018) and keypoints (Cao et al., 2017), which can be daunting for non-professional users. It becomes clear that there is a pressing need for the community to explore more accessible and less resource-intensive methods, with the effort of less compromise on the quality of image generation—a domain that, to our knowledge, remains uncharted to date.

In marked contrast, our proposed DiffusionTrend offers a streamlined, lightweight training approach that circumvents the need to train a diffusion model, thus liberating it from the reliance on expensive and resource-intensive computational infrastructure. This technique simplifies the workflow by dispensing with the requirement for intricate segmentation, pose extraction, and other preliminary processing steps for the input images, leading to a more accessible and economical solution. Specifically, a lightweight, compact CNN is utilized to outline the clothing in both the model and garment images. It harmonizes image and textual features and conducts clustering at the feature level to produce effective masks. Subsequently, we make full use of the latents derived from DDIM inversion (Song et al., 2020), which are replete with prior information and act as superior conduits for the detailed features of the garment. During the early stages of the diffusion denoising process, the seamless integration of the target garment into the model's image reconstruction is achieved by blending the latent representations of both the model and the garment. In the subsequent stages of denoising, leveraging the self-repairing capabilities of the pre-trained latent diffusion model, we maintain the model's identity and background coherence by selectively replacing the latents in the background areas. This approach ensures that the garment is harmoniously merged into the overall image, preserving the integrity of the original scene. Figure 1 shows several examples.

Figure 1: Images generated by the proposed DiffusionTrend model, given an input target model and a try-on clothing item both from DressCode dataset (Morelli et al., 2022).

Despite its suboptimal metric performance, our initial exploration with DiffusionTrend has yielded several valuable contributions that could indeed pave the way for further research and refinement in the realm of untrained diffusion models for virtual try-on technology:

- It eliminates the necessity for resource-intensive training of diffusion models on extensive datasets, thereby reducing the computational demands.
- It removes the need for cumbersome and user-unfriendly model inputs, streamlining the process for non-professional end-users.
- It delivers a visually appealing virtual try-on experience, highlighting the potential of diffusion models that do not require training for future research within the community.

These contributions underscore the significance of our DiffusionTrend model as a foundation for future advancements in virtual try-on technology, emphasizing the benefits of a training-free approach.

## 2 RELATED WORK

### 2.1 IMAGE EDITING THROUGH DIFFUSION PROCESSES

The practice of integrating specific content into a base image to produce realistic composites is prevalent in image editing leveraging diffusion processes. Initially, the field was dominated by text-based models for image editing (Brooks et al., 2023; Kawar et al., 2023). InstructPix2Pix (Brooks et al., 2023) employed paired data to train diffusion models capable of generating an edited image from an input image and a textual instruction. Conversely, Imagic (Kawar et al., 2023) harnessed a pre-trained text-to-image diffusion model to generate text embeddings that align with both the input image and the target text. The abstract nature of text poses a limitation in accurately delineating the subtleties of objects, therefore, image conditioning was introduced to offer more concrete and precise descriptions. DCFF (Xue et al., 2022) pioneered the use of pyramid filters for image composition, which was subsequently advanced by Paint by Example (Yang et al., 2023), employing CLIP embeddings of the reference image to condition the diffusion model. The majority of contemporary methodologies, such as Dreambooth (Ruiz et al., 2023) (all model parameters), Textual Inversion (Gal et al., 2022) (a word vector for novel concepts), and Custom-Diffusion (Kumari et al., 2023) (cross-attention parameters), relied heavily on fine-tuning techniques. In contrast, a handful of approaches (Hertz et al., 2022; Cao et al., 2023) adopted a training-free paradigm. Prompt-to-prompt (Hertz et al., 2022) modified the input text prompt to steer the cross-attention mechanism for nuanced image editing, while MasaCtrl (Cao et al., 2023) employed a mask-guided mutual attention strategy for non-rigid image synthesis and editing. These training-free methods offer a cost-effective alternative, eliminating extensive training while still delivering commendable generative outcomes.

### 2.2 VIRTUAL TRY-ON WITH DIFFUSION MODELS

Diffusion models have demonstrated remarkable efficacy in the domain of image editing, with image-based virtual try-on representing a specialized subset of these tasks, contingent upon a specific garment image. Adapting text-to-image diffusion models to accommodate images as conditions, is straightforward, but the spatial discrepancies between the garment and the subject's pose challenge the fidelity of texture details in the virtual try-on outcomes (Li et al., 2023; Gou et al., 2023; Morelli et al., 2023). Methodologies such as WarpDiffusion (Li et al., 2023), DCI-VTON (Gou et al., 2023),

and LADI-VTON (Morelli et al., 2023) conceptualize clothing as pseudo-words, employing warping techniques through CNN networks to adjust clothing to various poses, yielding satisfactory results. TryOnDiffusion (Zhu et al., 2023) employs a dual U-Net architecture for the virtual try-on task, implicitly conducting garment warping through the interplay between cross-attention layers. This approach effectively resolves the issue of texture misalignment without the need for a dedicated warp module. Similarly, the StableVITON (Kim et al., 2023) incorporates zero cross-attention blocks to condition the intermediate feature maps of a spatial encoder, thereby circumventing the requirement for a warp module. The Wear-Any-Way (Chen et al., 2024) enhances the process of virtual garment alteration, providing more adaptable control over the manner in which clothing is depicted. The IDM-VTON (Choi et al., 2024) enhances the virtual try-on process by integrating attention mechanisms and high-level semantic encoding into the diffusion model framework, ensuring an authentic clothing representation in virtual environments.

## 3 METHODOLOGY

### 3.1 PRELIMINARIES

**Latent Diffusion Models.** Latent diffusion models (LDMs) use an encoder $\mathcal{E}$ to convert an input image $x_0 \in \mathbb{R}^{H \times W \times 3}$ into a lower-dimensional $z_0 = \mathcal{E}(x_0) \in \mathbb{R}^{h \times w \times c}$. Here, the downsampling ratio is $f = H/h = W/w$, and $c$ is the channel number. The forward diffusion process is:

$$z_t = \sqrt{\bar{\alpha}_t} z_0 + \sqrt{1 - \bar{\alpha}_t} \epsilon, \quad \epsilon \sim \mathcal{N}(0, I), \tag{1}$$

where $\{\alpha_t\}_{t=1}^T$ denotes variance schedules, with $\bar{\alpha}_t = \prod_{i=1}^t \alpha_i$. A U-Net $\epsilon_\theta$ refines noise estimation. This is crucial for reconstructing the latent representation $z_0$ from the initial noisy state $z_T \sim \mathcal{N}(0, \mathbf{I})$:

$$z_{t-1} = \sqrt{\frac{\alpha_{t-1}}{\alpha_t}} z_t + \left( \sqrt{\frac{1}{\alpha_{t-1}} - 1} - \sqrt{\frac{1}{\alpha_t} - 1} \right) \cdot \epsilon_\theta \big( z_t, t, \tau_\theta(\mathcal{P}) \big). \tag{2}$$

The text encoder $\tau_\theta(\mathcal{P})$ converts text prompt $\mathcal{P}$ into an embedding that is integrated with the U-Net's intermediate noise representation using cross-attention mechanisms. At time step $t = 0$, the decoder $\mathcal{D}$ transforms the latent space representation $z_0$ back into the original image domain $x_0 = \mathcal{D}(z_0)$.

**DDIM Inversion.** DDIM inversion uses DDIM sampling (Song et al., 2020) to ensure deterministic sampling by setting the variance in Eq. (2). It assumes the reversibility of the ordinary differential equation (Chen et al., 2018) through incremental steps. This ensures a controlled transition from initial state $z_0$ to final noise $z_T$:

$$z_t^* = \sqrt{\frac{\alpha_t}{\alpha_{t-1}}} z_{t-1}^* - \sqrt{\frac{\alpha_t}{\alpha_{t-1}}} \left( \sqrt{\frac{1}{\alpha_{t-1}} - 1} - \sqrt{\frac{1}{\alpha_t} - 1} \right) \cdot \epsilon_\theta \big( z_{t-1}^*, t-1, \tau_\theta(\mathcal{P}) \big), \tag{3}$$

We start with the noisy latent state $z_T^*$ and proceed with denoising as outlined in Eq. (2). This method approximates $z_0^*$, which closely resembles the original latent representation $z_0$. Our goal is to incorporate information from a garment image $I^g$ into the reconstruction of the model image $I^m$, depicting the model wearing the garment from $I^g$.

To address the challenge of extensive training overhead and various model inputs, we choose not to alter any weights or structures of the pre-trained diffusion model. Instead, we introduce a lite-training visual try-on method called "DiffusionTrend". Our methodology consists of three stages. 1) A lightweight and compact CNN accurately delineates the apparel in both the model and garment images. 2) At an appropriate point in the process, garment details are integrated into the reconstruction phase of the model image. 3) To ensure the coherence of the generated background with the model, we use a latent substitution technique to restore the background, leveraging the diffusion model's restorative properties to blend it seamlessly with the newly rendered apparel. A comprehensive discussion of the first stage is in Sec. 3.2, while the latter two stages are discussed in Sec. 3.3.

### 3.2 PRECISE APPAREL LOCALIZATION

In conventional GAN-based (Lee et al., 2022; Ge et al., 2021; Xie et al., 2023) or current Diffusion-based (Zhu et al., 2023; Gou et al., 2023; Morelli et al., 2023) try-on methods, a precise clothing

Figure 2: Visual comparison of mask generation and try-on results. (a) Precision of apparel localization masks generated by our network compared to masks from attention maps. (b) Quality comparison of generation results: Reference U-Net method *vs.* DiffusionTrend.

mask is crucial. This mask ensures correct apparel placement on the model and accurate extraction of garment features while avoiding interference from non-garment regions. Traditional segmentation models (He et al., 2017; Kirillov et al., 2023) are typically employed to generate masks. However, in environments where numerous users engage in online consultations simultaneously, computational efficiency becomes crucial. For instance, Segment Anything (Kirillov et al., 2023), while effective, incurs a significant computational cost (93.74M parameters, 372 GFLOPs per $768 \times 1024$ image, 2.7G GPU memory), which can be impractical for real-world applications, especially when scalability and cost-effectiveness are paramount. Also, in our previous experiments, we attempted to extract masks using the intermediate attention map of a diffusion U-Net. During the inversion process, we utilized the concept of "clothes" to interact with image features via cross-attention. However, the masks extracted did not meet the precision requirements for virtual try-on tasks. As shown in Figure 2(a), the attention map roughly identifies the garment region but often includes extraneous parts.

Therefore, in consideration of balancing accuracy and saving computational resources, we have engineered a compact CNN in Figure 3(a) for precise apparel localization. By combining textual and visual features, we minimize manual intervention. Users only need to specify the target category (*e.g.*, upper garments, lower garments, or dresses), and the model automatically generates precise mask outputs. Our network accepts a model image $\mathcal{I}_0$ as input and processes it through three $3 \times 3$ $Conv_i^{3 \times 3}$ layers with $ReLU$ activation, culminating in the image features $\mathcal{I}_3$, as:

$$\mathcal{I}_i = ReLU\big(Conv_i^{3 \times 3}(\mathcal{I}_{i-1})\big), \quad i = 1, 2, 3. \tag{4}$$

An apparel-related prompt $\mathcal{P}$ is transformed into a text embedding $\mathcal{T}_0 = Clip(\mathcal{P})$ by a $Clip$ (Radford et al., 2021) text encoder with fixed parameters. The text embedding $\mathcal{T}_0$ is then advanced through two $FC_i$ layers with a $ReLU$ function in between, to produce the text-derived features $\mathcal{T}_2$ as:

$$\mathcal{T}_2 = FC_2\Big(ReLU\big(FC_1(\mathcal{T}_0)\big)\Big). \tag{5}$$

Next, we amalgamate the image feature $\mathcal{I}_3$ with the text feature $\mathcal{T}_2$ using a $1 \times 1$ $Conv^{1 \times 1}$ layer, followed by $Sigmoid$ and $Upsample$ functions as:

$$\mathcal{M} = Upsample\Big(Sigmoid\big(Conv^{1 \times 1}(\mathcal{I}_3 \cdot \mathcal{T}_2)\big)\Big). \tag{6}$$

For the training, we compute the $\ell_1$ norm between $\mathcal{M}$ and ground-truth mask $\mathcal{M}_{GT}$, employing this as the loss function to refine the network's parameters. Note that the predicted $\mathcal{M}$ serves as a mask for all clothing items on the model, which may encompass both upper and lower garments.

To delineate between the upper and lower garments and to extract their respective masks, we perform 2-clustering upon the masked features $\mathcal{M} \cdot \mathcal{I}_3$, which allows us to discern the top and bottom masks:

$$\mathcal{M}_{up}, \ \mathcal{M}_{low} = Cluster(\mathcal{M} \cdot \mathcal{I}_3). \tag{7}$$

It should be noted that for models adorned in addresses or when processing a garment image, $\mathcal{M}$ is utilized directly as the mask, thereby obviating the need for clustering.

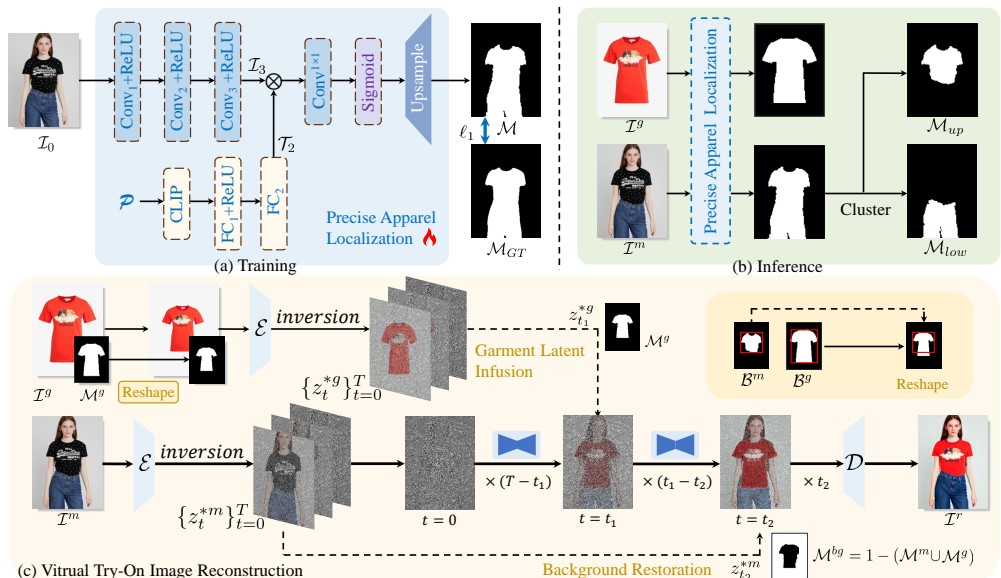

Figure 3: DiffusionTrend framework. (a) A lightweight apparel localization network to predict precise garment masks. (b) The network inference. (c) The reconstruction of the virtual try-on image.

Our apparel localization network is notably lightweight, making the training process highly cost-effective when compared to methods (Li et al., 2023; Kim et al., 2023; Gou et al., 2023) that require training a diffusion model. It takes us only 20 hours to process the entire DressCode dataset (Morelli et al., 2022) using two RTX 3090 GPUs. The localization network in Figure 3(b) operates fully automatically with only 2.00 GFLOPs and 0.15G of memory, and its accuracy analysis can be found in Sec. C of the Appendix.

## 3.3 VIRTUAL TRY-ON IMAGE RECONSTRUCTION

Initially, we adhered to MasaCtrl (Cao et al., 2023), using a reference U-Net to extract garment features and facilitating key-value exchange with the main U-Net during the attention stage, all without the need for additional training. Despite employing a dual-branch U-Net and attempting to guide the attention interaction with masks derived from our apparel localization network, the outcomes were less than satisfactory. This was especially true when it came to the accurate rendering of fine-grained garment details, as depicted in Figure 2(b). We surmise that this direct feature injection, while adept at generating content at the semantic level, falls short in capturing the fine-grained clothing details essential for virtual try-on applications.

Moreover, recent studies (Choi et al., 2024; Xu et al., 2024) indicate that employing U-Net for feature extraction necessitates treating it as a high-parameter module, which entails considerable training expenses. This is counterintuitive to our objective of achieving a solution that does not require training of the diffusion model. Consequently, we have been refining our methodology iteratively. As illustrated in Figure 3(c), our virtual try-on image reconstruction process encompasses two critical stages: the infusion of garment latents and the restoration of the background.

**Garment Latent Infusion.** Given a model image $\mathcal{I}^m$ and a garment image $\mathcal{I}^g$, we utilize the apparel localization network from Sec. 3.2 to acquire the masks $\mathcal{M}^m$ and $\mathcal{M}^g$, respectively. Next, we compute the bounding boxes $\mathcal{B}^m$ and $\mathcal{B}^g$ for $\mathcal{M}^m$ and $\mathcal{M}^g$, respectively. By reshaping the contents of $\mathcal{B}^g$ to the size of $\mathcal{B}^m$ and applying appropriate perspective transformations to simulate image rotation, we align the target garment with the position and size of the clothing in the model image.

The model image $\mathcal{I}^m$ and the aligned garment image $\mathcal{I}^g$ are then transformed into the latent representations $z_0^m = \mathcal{E}(\mathcal{I}^m)$ and $z_0^g = \mathcal{E}(\mathcal{I}^g)$. These latents are then subjected to the DDIM Inversion process as outlined in Eq. (3), yielding noisy latent sets $\{z_t^{*m}\}_{t=0}^T$ and $\{z_t^{*g}\}_{t=0}^T$.

Initiating from $z_T^{*m}$, we progressively reconstruct the model image using Eq. (2) where the latent at the $t$-th time step is $z_t^m$. Our goal is to decode an image of the model wearing the garment shown in image $\mathcal{I}^g$, achieving a harmonious fusion of the model's appearance with the desired attire. A literature review (Wu et al., 2023) has shown that the latent derived from the inversion process is rich in prior information, making it ideal for capturing the intricate features of the target garment. An effective strategy is to integrate the garment latent $z_{t_1}^{*g}$, preserved during the inversion stage, into the core of the garment mask $\mathcal{M}^g$ at time step $t_1$. This occurs early in the denoising process as:

$$z_{t_1}^* = z_{t_1}^m \cdot (1 - \mathcal{M}^g) + z_{t_1}^{*g} \cdot \mathcal{M}^g. \tag{8}$$

The denoising then continues with the infused latent. This step ensures the seamless information transfer from the garment image to the model image, achieving a precise attire on the model.

**Background Restoration.** While the garment latent infusion yields significant results, it negatively impacts the generation of background content in subsequent denoising steps. As shown in Sec. B of the Appendix, issues such as altered background color, distorted facial features, and unintended changes to other parts of the model's attire occur. To address this, we must implement strategies to restore the background and preserve the model's identity and background information.

Motivated by this, we inject the model latent $z_{t_2}^m$ into the regions outside the model clothing mask $\mathcal{M}^m$ at time step $t_2$, a later stage in the diffusion denoising process focused on generating detailed information. Leveraging the diffusion model's inherent repair capability, a few subsequent denoising steps integrate the background latents with the target garment. This process can be formalized as:

$$z_{t_2}^* = z_{t_2}^m \cdot \mathcal{M}^m + z_{t_2}^{*m} \cdot (1 - \mathcal{M}^m). \tag{9}$$

After extensive experiments, we found that using $\mathcal{M}^m$ to differentiate between the foreground and background is not optimal. As shown in the Sec. B of the Appendix, if the original model is wearing a short-sleeved garment and the target garment is long-sleeved, the sleeve in the generated image is incorrectly marked as background and replaced with the arm from the original model image $I^m$, causing a style mismatch. To solve this, we use the union of the model clothing mask $\mathcal{M}^m$ and the garment mask $\mathcal{M}^g$, and the complement as the background mask $\mathcal{M}^{bg}$, as expressed below:

$$\mathcal{M}^{bg} = 1 - (\mathcal{M}^m \cup \mathcal{M}^g). \tag{10}$$

Consequently, Eq. (9) is revised to the following:

$$z_{t_2}^* = z_{t_2}^m \cdot (1 - \mathcal{M}^{bg}) + z_{t_2}^{*m} \cdot \mathcal{M}^{bg}. \tag{11}$$

The subsequent denoising steps proceed on $z_{t_2}^*$ until the latent $z_0$ is reached according to Eq. (2). Ultimately, by decoding $z_0$, we obtain the generated try-on result image $\mathcal{I}^r$.

## 4 EXPERIMENTATION

### 4.1 EXPERIMENTAL SETUP

**Dataset.** We conduct extensive experiments on two high-resolution datasets from the VITON benchmark: VITON-HD (Choi et al., 2021) and DressCode (Morelli et al., 2022). The VITON-HD dataset includes 13,679 pairs of images, each featuring a front-view upper-body shot of women alongside corresponding in-store garments, split into 11,647 training pairs and 2,032 testing pairs. The DressCode dataset is larger, with 48,392 training pairs and 5,400 testing pairs, featuring front-view full-body images of individuals with corresponding in-store garments, categorized into upper-body, lower-body, and dresses.

To train the apparel localization network, we use model images from all DressCode. We extract relevant clothing portions from the DressCode training set label maps as our ground truth, $\mathcal{M}_{GT}$. We also integrate in-store garment images and their masks from the VITON-HD training set into $\mathcal{M}_{GT}$. For evaluation, we use the test sets from DressCode and VITON-HD to assess our method.

**Evaluation Metrics.** We assess outcomes in both paired and unpaired scenarios. In the paired scenario, the target human image and its corresponding garment image are used for reconstruction. In the unpaired scenario, different garment images are used for the virtual try-on experience. To evaluate the quality of images generated in the paired scenario, we use LPIPS (Zhang et al., 2018)

Table 2: Quantitative results on the VITON-HD and DressCode datasets.

| Method | VITON-HD | | | | DressCode | | | |
|---|---|---|---|---|---|---|---|---|
| | LPIPS↓ | SSIM↑ | FID↓ | KID↓ | LPIPS↓ | SSIM↑ | FID↓ | KID↓ |
| TryOnDiffusion (Zhu et al., 2023) | - | - | 13.447 | 6.964 | - | - | - | - |
| DCI-VTON (Gou et al., 2023) | 0.0530 | 0.8920 | 9.130 | 0.870 | 0.0443 | - | 11.800 | - |
| LaDI-VTON (Morelli et al., 2023) | 0.0910 | 0.8760 | 9.410 | 0.160 | 0.0640 | 0.9060 | 6.480 | 0.220 |
| WarpDiffusion (Li et al., 2023) | 0.0880 | 0.9850 | 8.610 | - | 0.0890 | 0.9010 | 9.187 | - |
| OOTDiffusion (Xu et al., 2024) | 0.0710 | 0.8780 | 8.810 | 0.820 | 0.0450 | 0.9270 | 4.200 | 0.370 |
| StableVITON (Kim et al., 2023) | 0.0732 | 0.8880 | 8.233 | 0.490 | 0.0388 | 0.9370 | 9.940 | 0.120 |
| IDM-VTON (Choi et al., 2024) | 0.1020 | 0.8700 | 6.290 | - | 0.0620 | 0.9200 | 8.640 | - |
| Wear-Any-Way (Chen et al., 2024) | 0.0780 | 0.8770 | 8.155 | 0.780 | 0.0409 | 0.9340 | 11.720 | 0.330 |
| DiffusionTrend (Ours) | 0.0919 | 0.8575 | 10.864 | 0.540 | 0.0721 | 0.9170 | 9.856 | 0.430 |

and SSIM (Wang et al., 2004) metrics to measure resemblance to the original image. In the unpaired scenario, we use FID (Heusel et al., 2017) and KID (Bińkowski et al., 2018) metrics to gauge the realism and fidelity of the synthesized images.

**Implementation Details.** We use the Adam optimizer (Kingma & Ba, 2014) for the apparel localization network with a learning rate of 1e-4, halving it every 10 epochs. The network is trained for 35 epochs on two RTX 3090 GPUs. We set the apparel-related prompt to "clothes" and apply our method to Stable Diffusion XL (SDXL) (Podell et al., 2023). For the inversion phase, we use an empty prompt, and for model image generation, the prompt is "model wearing clothes." We conduct DDIM sampling (Song et al., 2020) with 50 steps and set the classifier-free guidance to 7.5. Garment latent infusion occurs at time step $t_1 = 10$, and background restoration at $t_2 = 35$. The entire generation process is carried out on a single RTX 3090 GPU.

### 4.2 EXPERIMENTAL RESULTS

**Quantitative Results.** Table 2 shows the quantitative comparisons between DiffusionTrend and other methods on VITON-HD (Choi et al., 2021) and DressCode (Morelli et al., 2022) test datasets. The results on DressCode of DCI-VTON (Gou et al., 2023) and StableVITON (Kim et al., 2023) contain our implementation because no usable codes are given. Despite relying solely on pre-trained models for tasks like garment latent infusion and background restoration, our performance lags behind the state-of-the-art (SOTA) models. Past methods incur high training costs to ensure generated try-on images remain realistic and natural under various complex poses, as confirmed by evaluation metrics. In contrast, DiffusionTrend aims to provide a resource-efficient, user-friendly tool for quickly confirming purchase intentions under simple try-on poses. Without training the diffusion model on large datasets, our basic operations fall short in handling complex poses, leading to suboptimal performance. As the first visual try-on model without training on diffusion models, our approach differs from traditional methods in motivation and technical approaches, making traditional try-on dataset evaluations insufficient for comprehensively measuring our method's effectiveness. Thus, quantitative experiments serve only as a reference, while qualitative experiments will further demonstrate our superiority.

**Qualitative Results.** Figure 4 provides the qualitative comparison of DiffusionTrend with the state-of-the-art baselines on the VITON-HD (Choi et al., 2021) and DressCode (Morelli et al., 2022) datasets. The results indicate that our DiffusionTrend performs as well as baseline methods under simple poses.

First, most baseline methods fail to generate realistic wrinkles after applying warp techniques, simply transferring wrinkles from in-store garment images.

Second, our approach extracts richer detail features from noise latent, allowing for more accurate detail reconstruction in complex garment patterns. For instance, in the first row of Figure 4, the cartoon pattern on the garments reconstructed by our method more closely resembles the original image compared to other baseline methods. In the third row, while most methods erroneously reconstruct the garment as a short skirt, our approach accurately captures the appropriate skirt length and intricately details the metallic embellishments at the waist.

Finally, our method is not limited by the training dataset and works well on different types of images. Additional results are provided in Sec. A of the Appendix due to space constraints.

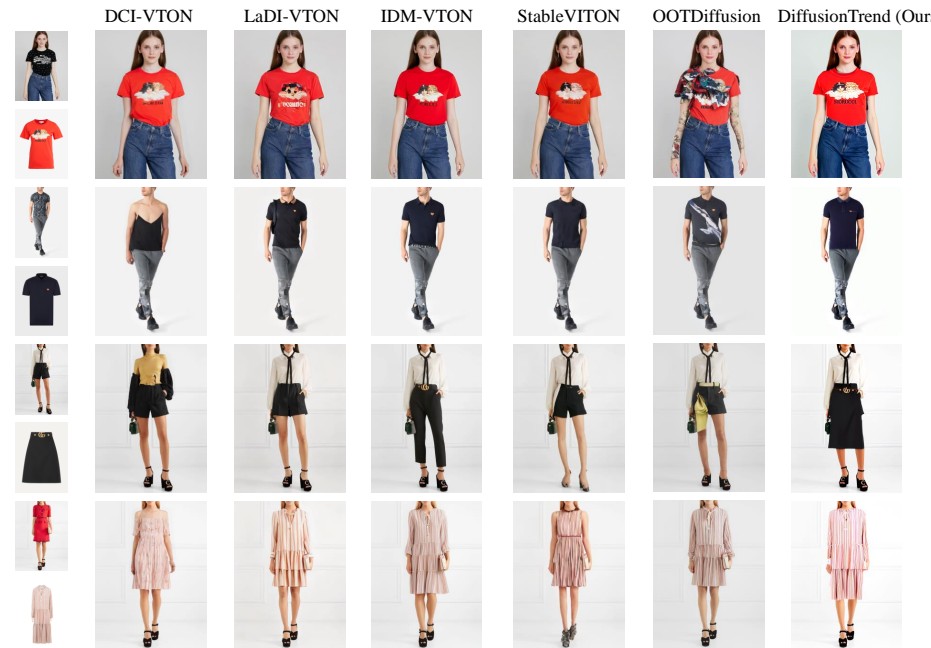

Figure 4: Qualitative comparisons on VITON-HD (Choi et al., 2021)(1st row) and DressCode (Morelli et al., 2022) (2nd ∼ 4th row).

### 4.3 ABLATION STUDY

In this section, we examine the optimality of our method's components and experimental settings, including background restoration and the timestep for latent infusion or replacement in Sec. 3.3. Quantitative results and qualitative results are presented in Table 3 and Figure 5, respectively.

$t_1$ **for Garment Latent Infusion.** The upper part of Table 3 shows that incorporating clothing information too early tends to lower the LPIPS, SSIM, and FID scores. Specifically, as observed in the top row of Figure 5, prematurely incorporating garment information results in a deficiency of integration between the model and the garment, effectively reconstructing their respective latent representations in disparate areas without any interaction. This issue is clearly visible, as there are pronounced boundaries between the clothing and the human figure, creating an impression of disjunction rather than a cohesive unity. On the other hand, introducing the information at a later stage impedes the development and enhancement of the garment's finer details. This is observable in the figure, where the cartoon patterns are prone to becoming indistinct, thereby compromising the overall quality of the image.

Table 3: Quantitative ablations for $t_1$ in Garment Latent Infusion and $t_2$ in Background Restoration.

| Timesteps | | Metrics | | | |
|---|---|---|---|---|---|
| $t_1$ | $t_2$ | LPIPS↓ | SSIM↑ | FID↓ | KID↓ |
| 0 | | 0.0804 | 0.9018 | 10.1231 | 0.44 |
| 5 | | 0.0763 | 0.9036 | 9.9633 | 0.44 |
| 10 (Ours) | 35 | 0.0731 | 0.9049 | 9.9322 | 0.44 |
| 15 | | 0.0722 | 0.9053 | 10.0390 | 0.46 |
| 20 | | 0.0709 | 0.9060 | 10.2479 | 0.47 |
| 25 | | 0.0701 | 0.9064 | 10.4147 | 0.48 |
| | 25 | 0.0765 | 0.9024 | 9.4499 | 0.41 |
| | 30 | 0.0748 | 0.9037 | 9.6527 | 0.43 |
| 10 | 35 (Ours) | 0.0731 | 0.9049 | 9.9322 | 0.44 |
| | 40 | 0.0735 | 0.9050 | 10.2813 | 0.46 |
| | 45 | 0.0736 | 0.9050 | 10.9275 | 0.51 |
| | 50 | 0.1697 | 0.8404 | 16.1841 | 0.76 |

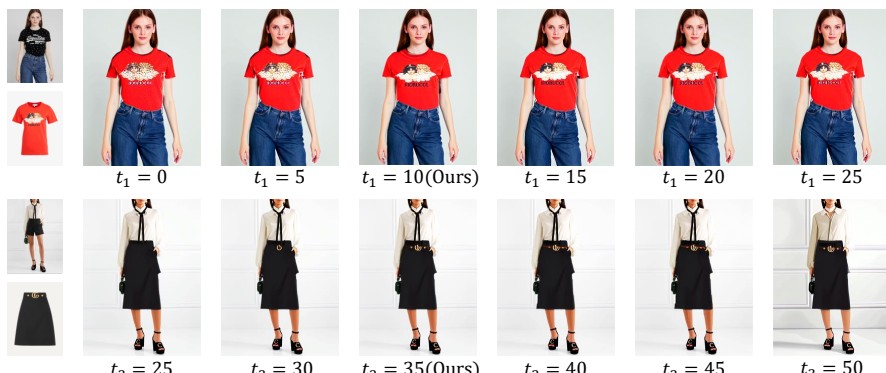

$t_1 = 0$    $t_1 = 5$    $t_1 = 10(Ours)$    $t_1 = 15$    $t_1 = 20$    $t_1 = 25$

$t_2 = 25$    $t_2 = 30$    $t_2 = 35(Ours)$    $t_2 = 40$    $t_2 = 45$    $t_2 = 50$

Figure 5: Visual ablations for $t_1$ in Garment Latent Infusion and $t_2$ in Background Restoration.

$t_2$ **for Background Restoration.** Although the quantitative results in Table 3 show that the FID and KID scores are higher at $t_2 = 25$ and 30, it can be observed from the second row of Figure 5 that, performing background restoration too early can negatively impact the generation of garment details. For instance, at $t_2 = 25$, the waist's metallic embellishments are lost during the reconstruction process; at $t_2 = 30$, the situation improves slightly, but the metallic dots on either side are still missing. In contrast, performing it too late creates a distinct boundary between the background and the foreground, resulting in unnatural outcomes, and leads to the over-rendering of details. In the figure at $t_2 = 45$, there are small red artifacts visible around the metallic dots.

## 5 LIMITATIONS AND FUTURE WORK

Our DiffusionTrend model encounters several challenges: (1) The refinement in clothing generation is not always optimal, occasionally leading to minor color variations and the presence of unnatural textures and patterns; (2) It encounters difficulties in generating complex poses, especially in rendering body parts that are not visible in the original model image, due to limitations inherent in the pre-trained model. For instance, if the original model is depicted wearing a long-sleeved shirt and the target garment is a short-sleeved one, the model is unable to convincingly render the exposed arms.

These limitations suggest that future research should prioritize enhancing the recovery of fine-grained details in clothing and strengthening the model's ability to generate previously unseen body parts. Moreover, further exploration into managing more complex poses and refining the quality of background restoration are areas that merit deeper investigation.

Despite these shortcomings, DiffusionTrend offers a low-cost, lightweight paradigm for the virtual try-on field that circumvents the need for extensive diffusion model training. Optimism remains high that this approach will continue to evolve as the field advances. We respectfully ask the academic community to recognize the value of this exploratory work and extend the necessary patience and support for its further development.

## 6 CONCLUSION

In this paper, we have introduced DiffusionTrend, a novel try-on methodology that forgoes the need for training diffusion models, thereby offering straightforward, conventional pose virtual try-on services with minimal computational demands. Capitalizing on sophisticated diffusion models, DiffusionTrend harnesses latents brimming with prior information to encapsulate the nuances of garment details. Throughout the diffusion denoising process, these details are effortlessly merged into the model image generation, expertly directed by a precise garment mask generated by a lightweight and compact CNN. Differing from other approaches, DiffusionTrend sidesteps the necessity for labor-intensive training of diffusion models on extensive datasets. It also dispenses the need for various types of user-unfriendly model inputs. Our experiments demonstrate that, despite lower metric performance, DiffusionTrend delivers a visually convincing virtual try-on experience, all while maintaining the quality and richness of fashion presentation.

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

APPENDIX

# A  ADDITIONAL QUANTITATIVE RESULTS

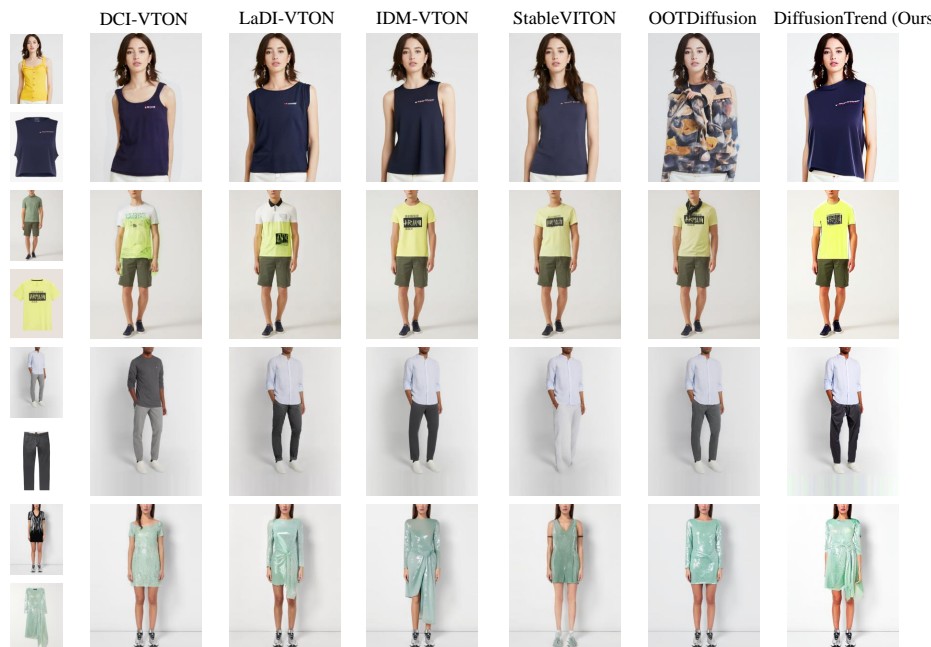

Figure 6:    Qualitative comparisons on VITON-HD (Choi et al., 2021)(1st row) and Dress-Code (Morelli et al., 2022) (2nd ∼ 4th row) datasets.

# B  ABLATION STUDIES ON $\mathcal{M}^{bg}$ AND BACKGROUND RESTORATION

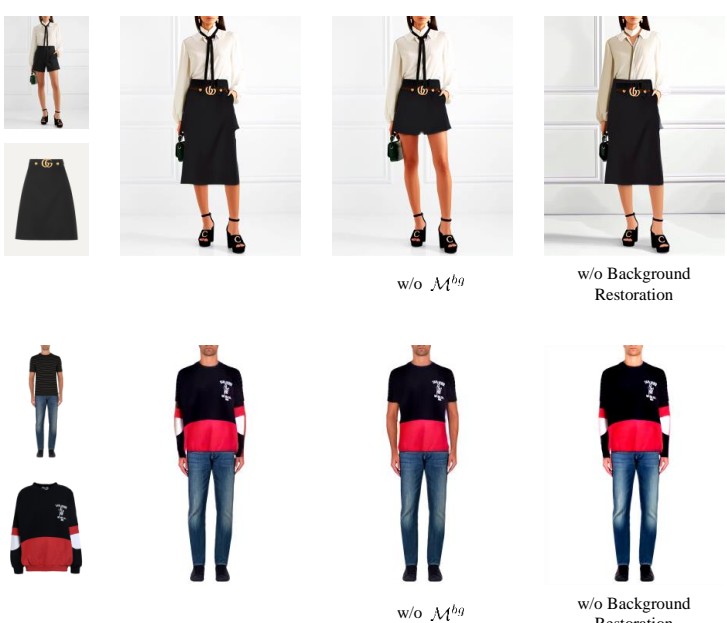

Figure 7:  Ablation studies on $\mathcal{M}^{bg}$ and background restoration are presented. Omitting $\mathcal{M}^{bg}$ results in inaccuracies such as the reconstruction of long skirts as short skirts and long sleeves as short sleeves. Without background restoration, various issues arise, including altered background colors, distorted facial features, and unintended changes to other parts of the model's attire.

## C  ACCURACY OF THE PRECISE APPAREL LOCALIZATION NETWORK

Table 4: Quantitative metrics on virtual try-on datasets for validating mask prediction accuracy.

| Metrics | DressCode | | | VITON-HD |
| --- | --- | --- | --- | --- |
| | upper_body | lower_body | dresses | |
| IoU | 0.7213 | 0.7138 | 0.7969 | 0.7357 |
| Dice | 0.8162 | 0.8038 | 0.8538 | 0.8194 |

To verify the effectiveness of our proposed precise apparel localization network, we conduct experiments on the test sets of the DressCode and VITON-HD datasets. For the DressCode dataset, we utilize label maps to extract ground truth masks, while for the VITON-HD dataset, we use image-parse-v3. In both cases, we extracted specific colored pixels and converted them into binary masks. We then use our apparel localization network to extract clothing masks and evaluated the performance using IoU (Intersection over Union) and Dice (Dice Coefficient) metrics. Generally, a model with an IoU above 0.5 and a Dice score above 0.7 is considered applicable in the research field.

Our results, as presented in Table 4, demonstrate that our compact CNN achieves high accuracy in mask extraction tasks, with IoU scores reaching 0.7 and Dice scores exceeding 0.8. These results validate the effectiveness and practicality of our approach.

## D  ANALYSIS OF DDIM INVERSION

Our proposed DiffusionTrend's performance is significantly influenced by DDIM inversion results. In our experiments, we find that the evolution of more powerful diffusion models is expected to yield better results with the same inversion method, as shown in the figure below. Moreover, we are encouraged by the ongoing research and development in the field of inversion methods, such as NULL-Text Inversion (Mokady et al., 2023), ReNoise (Garibi et al., 2024), Fixed-Point Iteration (Pan et al., 2023) and EasyInv (Zhang et al., 2024).

We believe that the emergence of better diffusion models and the potential for improvement in inversion methods will expand the application space for our training-free try-on paradigm.

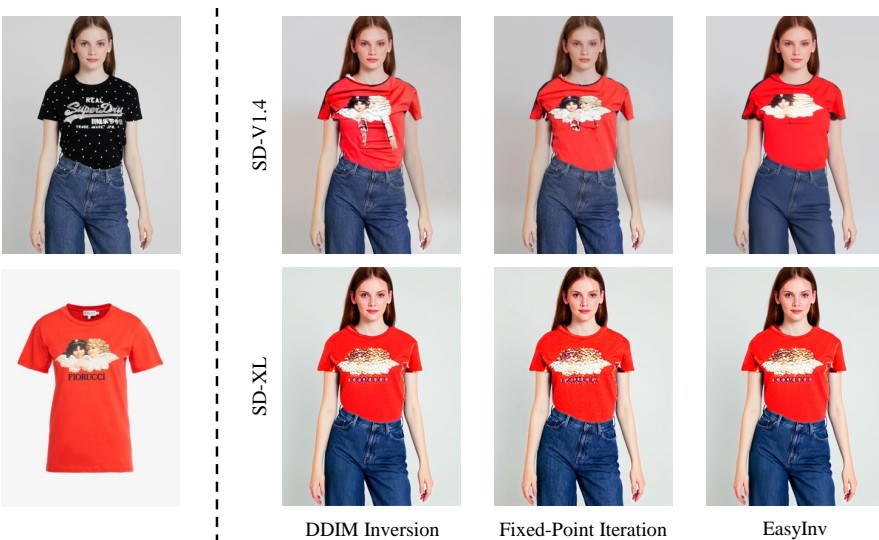

Figure 8: Improved inversion results achieved by SD-XL and advanced inversion methods.

# E   DISCUSSION ON WARPING GARMENTS USING PERSPECTIVE TRANSFORMATIONS

How to warp garments from their in-store flat state to a specific human pose is a critical part of the virtual try-on task. A natural thought is to use the current mature explicit warp module, which is also the initial direction of our approach.

We attempt to use the warp module from DCI-VTON (Gou et al., 2023) for pre-processing garment images but achieve poor results, as shown in Figure 9. The warped garment images are incomplete, leading to poor inversion and severely distorted reconstructions.

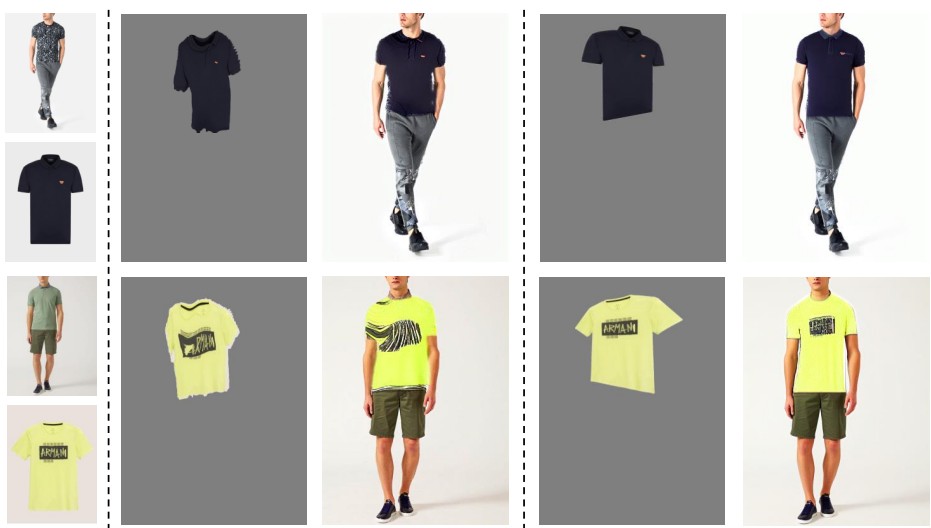

Explicit Warping                    Perspective Transformations (Ours)

Figure 9: Comparison of results using explicit warping methods and perspective transformations. Left: garment with warped / perspective transformation; Right: results generated using the processed garment.

We believe DCI-VTON's success relies on training the diffusion model on a large dataset, improving the compatibility between the warp module and the trained U-Net. In contrast, in our training-free method, using an independent warp module with an untrained U-Net does not produce satisfactory results. Additionally, starting the inversion process with incomplete warped images hinders reconstruction quality, further damaging the entire pipeline's output.

Therefore, we opt for perspective transformations to handle simple rotations and deformations, which can simulate basic poses. We acknowledge that perspective transformations cannot simulate the complex deformations that clothing undergoes in real try-on scenarios, as mentioned in our limitations section.

However, we believe this limitation does not negate our contribution to the field, which is the introduction of a training-free paradigm for virtual try-on, not requiring large-scale diffusion model training. In the future, we will explore other methods to improve garment deformation effects and enhance the usability of diffusion models for complex poses. For now, we call for the community to be open to innovative approaches and grant us patience and time for further development.

## F    MITIGATING POTENTIAL COLOR DISCREPANCIES

Some experimental results indicate a slight color discrepancy between the garment in our visual outputs and the original garment image. After extensive experimentation, we suggest mitigating this effect by appropriately reducing the sampling steps.

As demonstrated in Figure 10, when the sampling steps are set to 46, the generated clothing colors more closely match the original garment image. This adjustment significantly improves the visual fidelity of the results. However, we do not recommend reducing the sampling steps too much, as we also discover that excessive reduction introduces new issues. For instance, when the steps are reduced to 44, we observe distortion and inconsistency in the details of both the person and the background, which degrades the overall visual quality.

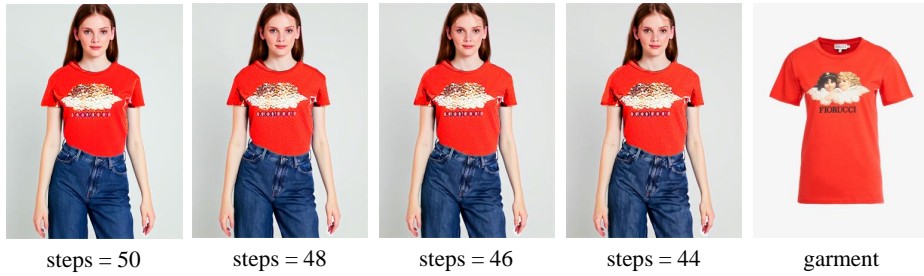

steps = 50            steps = 48            steps = 46            steps = 44            garment

Figure 10: Color oversaturation issue can be alleviated by reducing the sampling steps.

Additionally, some studies suggest that the color of generated images may be influenced by the classifier-free guidance scale. However, our attempts does not yield any significant improvements. Nevertheless, we will continue to explore other methods to further enhance the robustness of DiffusionTrend.

