# OpenReview forum: "DiffusionTrend: A Minimalist Approach to Virtual Fashion Try-On"
_ICLR.cc/2025/Conference — ICLR 2025 Conference Withdrawn Submission_

### Official Review · Reviewer_zUcf · 2024-10-18

**Soundness:** 1
**Presentation:** 3
**Contribution:** 1
**Rating:** 3
**Confidence:** 5

**Summary:**

This paper proposes a simple approach for VTON that does not require significant training in diffusion models. The main idea is to train a simple mask generator and replace the corresponding garment region in a given human image with a given garment image in the diffusion process. The experimental results are worse than the state-of-the-art methods.

**Strengths:**

(+) The writing is easy to follow.

**Weaknesses:**

(-) The motivations are defective. i) Most SOTA methods only fine-tune the diffusion models and are not too computationally extensive. ii) The densepose/segment map/clothes-agnostic representations/keypoints are not additional inputs, but intermediate results from existing tools, and therefore do not complicate use at all. iii) There are many existing human parsing models (no need to use SegmentAnything) that are powerful and lightweight, which can be obtained by googling the keyword "human parsing model".

(-) There is little technical novelty. The training of a simple segmentation network (Sec. 3.2) is trivial and as mentioned above, unnecessary. The editing through masking & copying strategy (Sec. 3.3) is also trivial in diffusion-based image editing (VTON is a niche of general image editing).

(-) The results are much worse than the state-of-the-art methods and are not meaningful. Specifically, due to the masking and replication strategies used in the proposed method, the clothes are not really "worn" on the body, but rather look like images attached to the body. For example, in Fig. 4, row 1: the wrinkles of the clothes are lost; row 3: the edge of the shorts are not removed and break the shape of the dress; row 4: the result ignores the body shape of the lady.

(-) The evaluation is limited. The types of clothes tested are much fewer than SOTA. All the examples have a simple background.

Therefore, the contributions of this paper are poor.

**Questions:**

Please see weaknesses above.

---

### Official Review · Reviewer_rmMS · 2024-10-22

**Soundness:** 3
**Presentation:** 3
**Contribution:** 2
**Rating:** 5
**Confidence:** 5

**Summary:**

The paper reviews the state of current virtual try-on technologies and identifies a key issue: high-quality and realistic image generation rely on resource-intensive models trained on extensive datasets, with existing diffusion-based methods involving up to 800 million parameters. This results in considerable training costs and computational demands that may deter users from trying on clothing with regular poses. Current methods also require diverse inputs that may be burdensome for non-professional users. In contrast, the paper introduces DiffusionTrend, a new approach that simplifies the virtual try-on process by forgoing extensive training on large datasets and avoiding complex preliminary input processing steps. The proposed method uses a lightweight CNN for feature-level clustering, producing effective masks without intensive training, and applies DDIM inversion to integrate detailed garment features into model images with background coherence.

**Strengths:**

1. The paper addresses the significant training cost issue of existing virtual try-on models and investigates low-cost methods for conducting try-ons, a valuable subject. The authors explore a viable approach to perform try-ons through the replacement of latents. Unfortunately, the method seems somewhat rudimentary, and there is room for improvement in the quality of the results.

2. The paper is well-articulated, and the data presented is credible.

**Weaknesses:**

1. The statement in line 161 "Traditional segmentation models (He et al., 2017; Kirillov et al., 2023) are typically employed to generate masks. However, in environments where numerous users engage in online consultations simultaneously, computational efficiency becomes crucial. For instance, Segment Anything (Kirillov et al., 2023), while effective, incurs a significant computational cost , which can be impractical for real-world applications, especially when scalability and cost-effectiveness are paramount" is inaccurate. In fact, input model images and input garment images can be automatically preprocessed through existing pretrained human-parsing models to obtain denpose, openpose, parsing images, agnostic images, etc. Please refer to the code [1][2]. Large segmentation model such as Segment Anything is NOT used. In addition, the computational overhead of image segmentation is very small compared to the diffusion denoising process.

2. There are many existing warping methods, yet this work has chosen to use a simple network to observe the mask and employ perspective transformations to deform the clothing. This does not simulate the myriad of clothing deformations that occur in a real try-on process, often leading to ill-fitting try-ons. In fact, since this work is divided into the warp cloth and diffusing process parts, it's necessary to show and discuss the effects of the warp cloth and beneficial to compare it with explicit warping models.

3. The inference speed of this method should be much slower than typical methods (possibly three times the duration), if I have not misunderstood, as the DDIM inversion takes nearly the same number of steps as the DDIM sampling.

4. The qualitative comparison in Fig. 4 appears to show that the results produced by DiffusionTrend have noticeably oversaturated colors.

[1] https://github.com/sangyun884/HR-VITON/issues/45

[2] https://github.com/levihsu/OOTDiffusion

**Questions:**

1. We observed background distortion and altered color without restoration. However, according to Eq. 8, the background and foreground are treated equally, does this mean that the clothing is equally likely to experience severe distortion? There is no discussion in the article concerning this issue.

**Details Of Ethics Concerns:**

No ethics review is needed.

---

### Official Review · Reviewer_gExg · 2024-11-01

**Soundness:** 1
**Presentation:** 1
**Contribution:** 2
**Rating:** 1
**Confidence:** 4

**Summary:**

This paper proposes a virtual try-on method which includes a lightweight CNN to predict masks and then infusing this information with DDIM inversion.

**Strengths:**

The strength of this paper lies in their argument for a resource constrained setup to achieve virtual try-on.

**Weaknesses:**

In my humble opinion, the biggest drawback lies in the assumption this paper makes about not using accurate off-the-shelf segmentation methods. SDXL base model which is likely used in this paper has around 3.5 billion parameters, while a SAM-B [1] has around 94.7 million parameters and SAM2 [2] Hiera Tiny has around 38.9 million parameters. These segmentation methods are significantly less computationally intensive as the base diffusion model. In my understanding of the proposal here, the segmentation requires a single forward pass, the outputs of which is used in the different steps of the diffusion model.

The paper has several further drawbacks:
 - Using an off-the-shelf segmentation method bypasses training anything completely. Comparing the lightweight CNN against  SAM, SAM2 or SCHP [3] would help show the advantage of the proposal.
 - Comparing the accuracy of masks in Figure 2 is done against attention maps. It would be insightful to the reader if the comparison was against more reliable segmentation methods like SAM, SAM2 or SCHP inspite of their additional resource requirements.
 - Lacks specific details on many information (I have marked them as questions in the next sections).
   - Which method was used in 2-clustering mentioned in Eq. 7?
   - Reshaping of the masks shown on the top-right of in Figure 3c. Why and how is this reshaping done? It is mentioned in the paper that
> By reshaping the contents of Bg to the size of Bm and applying appropriate perspective transformations to simulate image rotation, we align the target garment with the position and size of the clothing in the model image.
It is unclear how this transformation was achieved. Why not use the M^m directly since thats the only place that needs to be replaced.
   - Following the previous question, M^g is used in Eq. 8 which is the garment mask. It seems unclear why M^g is used since it does not align with the model image.

[1] https://arxiv.org/abs/2304.02643
[2] https://arxiv.org/abs/2408.00714
[3] https://arxiv.org/abs/1910.09777v1

**Questions:**

-  It would be great if the authors could provide a comparative analysis of accuracy, computational efficiency, and resource requirements for their proposal of a lightweight CNN against off-the-shelf segmentation methods like SAM, SAM2 or SCHP.
- Many papers perform a user study as this shows the actual use case of whether customers like the images they are being shown by this method vs state-of-the-art. For e.g. the user study in MM-VTON [4] display results from different methods to the users and ask them to
> either select the best result or opt for “hard to tell.”

Additionally, users can also be asked for perceived realism of the results compared to other methods.
- I believe SSIM/LPIPS being slightly lower/higher is not an issue (in my humble opinion) as the main argument of this paper is resource-constrained setup. However, results need more backing. Specifically, comparisons to different off-the-shelf segmentation methods. A table on accuracy vs resource trade-offs with different setups (e.g. Precise Apparel Localization, attention maps, SAM, SAM2, SCHP ) will help the reader understand what makes their proposal more robust.

[4] https://arxiv.org/abs/2406.04542

---

### Official Review · Reviewer_uNho · 2024-11-08

**Soundness:** 3
**Presentation:** 3
**Contribution:** 2
**Rating:** 5
**Confidence:** 4

**Summary:**

The paper introduces a method for training-free virtual try-on. The input is a target image of a person and a source garment image, and the output should be the same target identity and environment, but with the new garment. The main insight is injecting features from the garment to the target image while it is being generated with a diffusion model.

**Strengths:**

Simple and efficient method
Mixes image-space deformation with neural generation, which is refreshing
Training free method

**Weaknesses:**

Results are underwhelming
Limited innovation
No user study
Can handle simple cases with simple background and lighting

**Questions:**

The paper claims that its main advantage is being training and optimization free. However, the proposed method is quite simple, and has many alternatives to try out. For example, using cross image attention, anydoor, and many others to pass appearance in the correct location.
There are many design choices that are not clearly justified.
Firstly, why train a simple segmentation model instead of using one off-the-shelf, or training an efficient network with a teacher-student approach from an off-the-shelf model? Indeed, there are many artifact that seem to be from a bad mask, such as residuals of the original source garments.
More importantly, the generation of the source garment is agnostic of the target environment, making it incompatible in terms of lighting etc. There is a short discussion on the matter, but as an adhoc solution that does not address the main issue.
Lastly, the source deformation is rather limited, meaning that different poses would be hard to satisfy. Experiments with different poses should be added, and compared to alternatives.

---

### Note · Authors · 2024-11-29

I have read and agree with the venue's withdrawal policy on behalf of myself and my co-authors.